# Study of Practical Analysis Method for Shear Warping Deformationof Composite Box Girder with Corrugated Steel Webs

**DOI:** 10.3390/ma16051845

**Published:** 2023-02-23

**Authors:** Maoding Zhou, Yuanhai Zhang, Pengzhen Lin, Wei Ji, Hongmeng Huang

**Affiliations:** 1Department of Civil Engineering, Gansu Agricultural University, Lanzhou 730070, China; 2College of Civil Engineering, Lanzhou Jiaotong University, Lanzhou 730070, China

**Keywords:** composite box girder, corrugated steel webs, shear warping deformation, simplified method, beam segment finite element

## Abstract

Shear warping deformation is an important part of the flexural and constrained torsion analysis of composite box girder with corrugated steel webs (CBG-CSWs), which is also the main reason for the complex force analysis of box girders. A new practical theory for analyzing shear warping deformations of CBG-CSWs is presented. By introducing shear warping deflection and corresponding internal forces, the flexural deformation of CBG-CSWs is decoupled to the Euler-Bernoulli beam (EBB) flexural deformation and the shear warping deflection. On this basis, a simplified method for solving shear warping deformation using the EBB theory is proposed. According to the similarity of the governing differential equations of constrained torsion and shear warping deflection, a convenient analysis method for the constrained torsion of CBG-CSWs is derived. Based on the decoupled deformation states, a beam segment element analytical model applicable to EBB flexural deformation, shear warping deflection, and constrained torsion deformation is proposed. A variable section beam segment analysis program considering the variation of section parameters is developed for CBG-CSWs. Numerical examples of constant and variable section continuous CBG-CSWs show that the stress and deformation results obtained by the proposed method are in good agreement with the 3D finite element results, verifying the effectiveness by the proposed method. Additionally, the shear warping deformation has a great influence on the cross-sections near the concentrated load and middle supports. This impact along the beam axis decays exponentially, and the decay rate is related to the shear warping coefficient of the cross-section.

## 1. Introduction

In recent years, composite box girders with corrugated steel webs (CBG-CSWs) have been widely used in the construction of medium- and long-span bridges because of their good mechanical properties and small dead weights. Corrugated steel web (CSW) can effectively improve the mechanical properties of the structure against buckling [1]. Additionally, the accordion effect of CSW makes it almost bear no axial bending moment or axial force, thereby improving the application efficiency of prestress [2,3,4]. Under bending load, CSWs and composite members [5] are similar in that they are anisotropic in plane. Therefore, the orthogonality of the web makes the classical EBB or Timoshenko beam theory no longer applicable to such structures. For this reason, scholars have been dedicated to developing simple analytical methods with high accuracy [6,7,8,9,10,11,12,13,14,15]. For example, Wu et al. [6] proposed a simplified analysis method of plane section assumption, which corresponded well to the classical EBB theory. On this basis, some researchers introduced shear additional angles to analyze the shear deformation of steel webs, so as to solve the deflection of composite girders [7,8,9]. These analytical methods can obtain the deflection of CBG-CSWs accurately, but they fail to consider the stress variation along the thickness of concrete flanges. To obtain the stress distribution of flanges, some researchers proposed a composite flexural mechanical model in which the flanges of the CBG-CSWs rotate around their respective centroids and the centers of the upper and lower flanges rotate around the centroid of cross-section [10,11,12]. The model is widely applied because it is more in line with the flexural characteristics of CBG-CSWs. Nie et al. [13], according to the flexural mechanical properties, put forward that the flexural deformation of CBG-CSWs is modeled as the combination of truss action and bending action. Based on the composite mechanical model, Chen et al. [14,15] proposed a beam segment analysis method and studied the influence of prestress, diaphragm. When the shear warping deformation of CWS is considered, solution accuracy can be higher, but the solution process will be extremely complicated and not suitable for engineering application. Therefore, how to simplify the calculation for shear warping deformation of CBG-CSWs is the key to facilitate engineering application.

As a special box girder, the torsional stiffness of CBG-CSWs is significantly weakened due to the accordion effect of CSWs [16]. The important form of restrained torsion is the shear warping deformation of the cross-section, which has been deeply studied by scholars for many years [17,18,19,20,21]. Murín and Choi et al. [17,18] respectively proposed a new beam theory to analyze the thin-walled members affected by shear warping deformation, and pointed out that the impact of shear warping deformation should not be ignored, especially for the closed cross-section. Based on the constrained torsion theory of thin-wall box girder, scholars analyzed the overall torsion behavior of CBG-CSWs by equivalent conversion of CWS [22,23,24,25]. These torsional analysis theories can obtain good solution accuracy, but the complex calculation process seriously hinders the wide application of these methods. To facilitate the solution, Andreassen and Cambronero-Barrientos et al. [26,27] proposed a high-order finite beam element method to analyze the shear deformation, bending-torsional deformation, and generalized warping displacement of components. Mokos and Diskaros et al. [28,29] introduced the boundary element method (BEM) into the analysis of flexure, torsion, and distortion of component. Compared with 3D finite element method (3D FEM), these numerical methods effectively simplify the modeling process of composite box girder and improve the solution efficiency, but still cannot quickly estimate the warping stress of the cross-section. Therefore, seeking a simple formula for calculating the warping deformation of composite box girder will help to improve the design efficiency of this type of bridge.

For the flexural analysis of the orthotropic CBG-CSWs, the traditional EBB cannot consider the shear deformation of the structure and the Timoshenko beam cannot obtain a more accurate stress distribution. The existing literature often couples the bending and shear deformation states, which makes the analytical process more difficult. In addition, the torsion solution process of composite box girder based on the torsion analysis theory of thin-walled box girder is relatively complex. The root causes for the difficulty in analyzing the complex flexural deformation and restrained torsional deformation of the CBG-CSWs is the shear warping deformation of box wall. However, it is often necessary to quickly estimate the impact of these shear warping deformations in practical engineering. In this paper, the flexural deformation of CBG-CSWs is firstly decoupled into two deformations, the EBB deformation and the shear warping deformation. Then, the relationship between the deflection of EBB and shear warping displacement is established according to the equilibrium condition of equal shear forces in cross-section. On this basis, a simplified analysis method of shear warping deformation is proposed. Subsequently, based on the similarity of the governing differential equations of constrained torsion and shear warping deformation, a simplified analysis method of CBG-CSWs constrained torsion is presented. Finally, a unified beam segment element model is proposed for decoupled deformations, and the accuracy of the proposed method is verified by numerical calculations.

## 2. Flexural Deformation Decoupling of CBG-CSWs

The analytical coordinate system and the cross-sectional size parameters of the typical CBG-CSWs are shown in Figure 1a,b, respectively. The flexural-torsional mechanical model of CBG-CSWs is analyzed under the following assumptions: (1) the vertical relative displacement between upper and lower flanges are ignored; (2) the shear lag effect of the upper and lower flanges is ignored, and each of them satisfy the plane assumption independently, but the whole section does not meet the plane assumption [10,11,12,13,14,15]; (3) the axial stiffness of the web is ignored; (4) the periphery of the cross section is rigid [25]; (5) the materials are all ideal elastic materials, and there is no relative slip between steel and concrete.

### 2.1. Flexural Displacement Function of CBG-CSWs

When subjected to vertical load, the CBG-CSWs will have bending deformation, as shown in Figure 2. If *w* is used to represent the vertical deflection, then according to assumption (2) the deflection angles of the upper and lower flanges of the CBG-CSWs are both *w*′(*z*). The angle around the *x*-axis of the line between the centers of the upper and lower flanges is *θ*(*z*), and the angle around the *x*-axis of the line between the top edge and the bottom of the CSWs is *α*(*z*). Under the combined influence of CSWs’ folding effect and shear deformation, the angles *w*′(*z*), *θ*(*z*) and *α*(*z*) are not equal [11,13]. According to Figure 1 and Figure 2, the longitudinal bending displacements *u*_u_(*y*,*z*) and *u*_l_(*y*,*z*) at any point of the upper and lower flanges can be respectively expressed as:(1)$$u_{u}(y,z)=h_{u}\theta (z)-(y+h_{u})w^{\prime }(z)$$
(2)$$u_{l}(y,z)=-h_{l}\theta (z)-(y-h_{l})w^{\prime }(z)$$

Through to Equations (1) and (2), the angle *α*(*z*) of the equivalent longitudinal displacement of the steel web around the *x*-axis can be obtained:(3)$$\alpha (z)=\frac{u_{u}(-a_{u},z)-u_{l}(a_{l},z)}{h_{w}}=\chi \theta (z)+(1-\chi )w^{\prime }(z)$$
where *χ* = *h*/*h*_w_. According to the condition of continuous deformation, the equivalent longitudinal displacement *u*_w_(*y*,*z*) of CSWs can be obtained as:(4)$$u_{w}(y,z)=-y\alpha (z)$$

### 2.2. Deflection Decoupling Based on Generalized Displacement

Due to the influence of shear warping deformation, the bending deformation of CBG-CSWs is quite different from the traditional Euler-Bernoulli beam displacement. As a result, the analytical process is complicated. Herein, the deflection *w*(*z*) of CBG-CSWs is decoupled into the deflection *w*_0_(*z*) of EBB conforming to the plane assumption and the deflection *f*(*z*) caused by shear warping deformation of web. Their deflective relationship is shown in Figure 3. By decoupling the deflection *w*(*z*) in Equations (1) and (2), the longitudinal bending displacement of the flanges can be obtained:(5)$$u(y,z)=-y[{w^{\prime }}_{0}(z)+f^{\prime }(z)]\;+\rho (y)[w^{\prime }(z)-\theta (z)]\;=-y{w^{\prime }}_{0}(z)-\xi (y)f^{\prime }(z)$$
where the value of *ρ*(*y*) is −*h*_u_ for the upper flange and *h*_l_ for the lower flange. *ξ*(*y*) is the distribution function of generalized longitudinal deflection on the upper and lower flanges. If *λf*′(*z*) = *w*′(*z*) − *θ*(*z*), *λ* is the coefficient to be calculated, then *ξ*(*y*) can be expressed as:(6)ξ(y)=y−λρ(y)

According to the displacement–strain relation and Hooke’s law, the flexural normal stress *σ*(*x*,*y*,*z*) of the flanges can be obtained from Equation (6).
(7)$$\sigma (y,z)=E_{c}\frac{\partial u}{\partial z}=-E_{c}yw_{0}^{\prime \prime }(z)-\;\;\;E_{c}\xi (y)f^{\prime \prime }(z)$$
where *E*_c_ is the elastic modulus of concrete. The first term at the right end of Equation (7) is the stress $\sigma_{E}$ of EBB, and the second term is the normal stress $\sigma_{\xi }$ caused by shear warping deformation, i.e.,
(8)$$\sigma_{\xi }=-\;\;\;E_{c}\xi (y)f^{\prime \prime }(z)$$

If the bending moment *M* of CBG-CSWs is only synthesized from EBB stress $\sigma_{E}$, then the warping stress $\sigma_{\xi }$ neither results in axial force on the whole cross-section nor bending moment with arm *y*, i.e.,
(9)$$\int_{A}\sigma_{\xi }dA=0$$
(10)$$\int_{A}\sigma_{\xi }ydA=0$$
where *A* is the area of CBG-CSWs section. Equation (8) is substituted into Equations (9) and (10). Equation (9) satisfies self-equilibrium, and Equation (11) can be obtained from Equation (10), and then the coefficient *λ* can be obtained.
(11)$$I_{x}=\lambda I_{y\rho }$$
where $I_{x}=\int_{A}y^{2}dA$ and represents the moment of inertia of the cross section, $I_{y\rho }=\int_{A}y\rho (y)dA$ and can be defined as the generalized product of inertia. Its parameter expression can be obtained by substituting *ρ*(*y*):(12)$$I_{y\rho }=h_{u}^{2}A_{u}+h_{l}^{2}A_{l}$$

The generalized moment *M*_ξ_ is defined as the synthesis of the shear warping stress with arm *ξ*, i.e.,
(13)$$M_{\xi }=\int_{A}\xi \sigma_{\xi }dA$$

Substituting Equation (8) into Equation (13) to get,
(14)$$M_{\xi }=-\;\;\;E_{c}I_{\xi }f^{\prime \prime }(z)$$
where $I_{\xi }=\int_{A}\xi^{2}dA$ and is defined as the generalized warping moment of inertia. Its parameter expression can be obtained from Equations (6), (11) and (12):(15)$$I_{\xi }=(\lambda -1)I_{x}$$

According to Equations (8) and (14), the warping normal stress $\sigma_{\xi }$ can be expressed as:(16)$$\sigma_{\xi }=\frac{M_{\xi }}{I_{\xi }}\xi$$

Therefore, the total flexural normal stress *σ* of CBG-CSWs can be given by:(17)$$\sigma =\frac{M}{I_{x}}y+\frac{M_{\xi }}{I_{\xi }}\xi$$

## 3. Simplified Method for Shear Warping Deformation of CBG-CSWs

### 3.1. Governing Differential Equation of Shear Warping Deformation

In the above analysis, the flexural deformation of CBG-CSWs has been decoupled to EBB-bending deformation and shear warping deformation. Thus, the deformation analysis of EBB is not detailed. The shear warping deformation can be analyzed by the energy variational principle (EVM) [30]. According to Equation (8), the warping normal strain $\varepsilon_{\xi }$ of the CBG-CSWs flanges can be expressed as,
(18)$$\varepsilon_{\xi }(y,z)=-\xi (y)f^{\prime \prime }(z)$$

According to Equation (3), the equivalent shear strain $\gamma_{yz}$ of the CSW be expressed as,
(19)$$\gamma_{yz}=\frac{\partial u_{w}}{\partial y}+\frac{\partial w}{\partial z}=\chi [w^{\prime }(z)-\theta (z)]=\lambda \chi f^{\prime }(z)$$

Then the strain energy of CBG-CSWs shear warping deformation can be expressed as follows:(20)$$U=\frac{1}{2}\int_{0}^{l}\int_{A}(E_{c}\varepsilon_{\xi }^{2}+G_{w}\gamma_{yz}^{2})dAdz\;=\frac{1}{2}\int_{0}^{l}E_{c}I_{\xi }f^{\prime \prime }{\left(z\right)}^{2}dz\;+\;\;\frac{1}{2}\int_{0}^{l}\lambda^{2}\chi^{2}G_{w}A_{w}f^{\prime }{\left(z\right)}^{2}dz$$
where *G*_w_ is the equivalent shear modulus of the CSW and with reference to relevant research [4,7], its expression is *G*_w_ = [(*a* + *b*)/(*a* + *c*)]*G*_s_. Here, *G*_s_ is the shear modulus of steel, and *a*, *b,* and *c* are waveform parameters, as shown in Figure 4. *A*_w_ is the area of CSWs.

Under the action of external load *p*(*z*), the potential energy of the CBG-CSWs can be expressed as,
(21)$$V=-\int_{0}^{l}p(z)f(z)dz$$

Based on the EVM and Equations (20) and (21), the governing differential equation of shear warping deflection *f* can be obtained:(22)$$f^{\left(4\right)}(z)-k_{\xi }^{2}f^{\prime \prime }(z)=\frac{p(z)}{E_{c}I_{\xi }}$$
where the superscript (4) is the fourth derivative of the function $k_{\xi }=\lambda \chi \sqrt{\frac{G_{w}A_{w}}{E_{c}I_{\xi }}}$ and is defined as the flexural shear warping coefficient. The boundary conditions corresponding to the governing differential equation of shear warping deflection can be obtained as [31,32]: for the fixed end, f=0 and $f^{\prime }=0$; for the simply supported end, f=0 and $f^{\prime \prime }=0$; for the free end, $f^{\prime \prime }=0$ and $f^{\prime \prime \prime }-k_{\xi }^{2}f^{\prime }=0$.

It can be seen from Equation (22) that this governing differential equation is in the same form as those of the constrained torsion [18]. The governing differential equation of the torsion of CBG-CSWs is expressed as [22,23],
(23)$$\phi^{\left(4\right)}(z)-k_{\bar{\omega }}^{2}\phi^{\prime \prime }(z)=\frac{\zeta m(z)}{{E^{\prime }}_{c}I_{\bar{\omega }}}$$
where φ is the torsion angle of the cross-section; $I_{\bar{\omega }}$ is the generalized principal sectorial inertia moment of the whole cross-section; *ζ* is the characteristic parameter of restrained torsion; *m*(*z*) is the torsional distribution load on CBG-CSWs; $k_{\bar{\omega }}=\sqrt{\frac{\zeta G_{c}I_{d}}{{E^{\prime }}_{c}I_{\bar{\omega }}}}$ and is defined as the torsional shear warping coefficient; *G*_c_ is the shear modulus of concrete; *I*_d_ is the torsional moment of inertia; $E_{c}^{\prime }$ is the corrected elastic modulus of concrete [25]. Therefore, the governing differential Equation (22) can be solved by the initial parameters method (IPM) from existing literature [25,31].

### 3.2. Simplified Analysis Method for Flexural Shear Warping

Since the shear stress of CSW can be approximately equivalent to a uniform distribution along the vertical direction of the web, the shear stress *τ*_1w_(*z*) can be analyzed by Equation (24) when the shear force of the whole cross-section is synthesized by the web. However, with the increase of bridge span, the thickness of concrete flange of composite box girder cannot be ignored relative to the height of the steel web [33]. If the influence of flange is considered, the shear stress *τ*_2w_(*z*) at any point of CSW can be expressed as Equation (25) [33,34,35].
(24)$$\tau_{1w}(z)=\frac{Q(z)}{2h_{w}t_{w}}$$
(25)$$\tau_{2w}(z)=\frac{Q(z)S_{c}}{2I_{x}t_{w}}$$
where *Q*(*z*) is the shear force; *S*_c_ is the area moment of the upper (or lower) flange against the *x*-axis of the section centroid; *t*_w_ is the thickness of CSW. The following relation can be approximated from Equation (19) and Equations (24) and (25):(26a)$${f^{\prime }}_{1}(z)=\frac{Q(z)}{2\lambda \chi G_{w}h_{w}t_{w}}=\frac{E_{c}I_{x}}{2\lambda \chi G_{w}h_{w}t_{w}}w_{0}^{\prime \prime \prime }=\eta_{1}w_{0}^{\prime \prime \prime }$$
(26b)$${f^{\prime }}_{2}(z)=\frac{Q(z)S_{c}}{2\lambda \chi G_{w}I_{x}t_{w}}=\frac{E_{c}S_{c}}{2\lambda \chi G_{w}t_{w}}w_{0}^{\prime \prime \prime }=\eta_{2}w_{0}^{\prime \prime \prime }$$

Obviously, Equations (26a) and (26b) establishes the relation between *w*_0_ and *f*, and the analysis and solution of *w*_0_ can be obtained by EBB theory. Through the analysis of Equations (26a) and (26b), the approximate expression of *f* can be obtained. The expressions of generalized warping normal stress and deflection can be obtained from Equations (8), (26a) and (26b) as follows:(27)$$\sigma_{\xi }(z)=-\eta_{i}\;\;\;E_{c}w_{0}^{\left(4\right)}(z)\xi =\eta_{i}\frac{p(z)}{I_{x}}\xi$$
(28)$$f(z)=-\eta_{i}w_{0}^{\prime \prime }(z)+C=\eta_{i}\frac{M(z)}{EI_{x}}+D_{1}z+D_{2}$$
where *D*_1_ and *D*_2_ are integration constants, whose values are determined by the boundary condition; the subscript of *η_i_* is set to 1 or 2. When the $w_{0}^{\left(4\right)}(z)$ exists, the $\sigma_{\xi }$ can be quickly obtained by Equation (27) and the deflection *f* of CBG with constant section can be obtained quickly by Equation (28). However, for the action point of concentrated load *P*, where the *P* should be equivalent to the distributed load *p*(*z*) (shown in Figure 5). This equivalent process has been demonstrated in the literature [30] and will not be described herein. Therefore, based on EBB results, the shear warping deflection and stress of CBG-CSWs can be quickly obtained by Equations (27) and (28).

### 3.3. A Simplified Analysis Method for Constrained Torsion

By comparing the Equations (16) and (27) of the normal stress in flexural shear warping deformation, it can be known that there must be a definite proportional relationship between the generalized moment *M*_ξ_(*z*) and the distributed load *p*(*z*). The generalized moment *M*_ξ_(*z*) of the cross-section is complicated to solve, while the distributed load *p*(*z*) can be obtained quickly. Thus, the relationship between *M*_ξ_(*z*) and *p*(*z*) is established, and $\sigma_{\xi }$ can be obtained from *p*(*z*). Similarly, according to the similarity between the governing differential Equations (22) and (23), *m*(*z*) can be used to solve the shear warping stress $\sigma_{\varpi }$ of CBG-CSWs constrained torsion. Taking a simply supported CBG under uniformly distributed load *p* (Figure 6) as an example, the relationship between *M*_ξ_ and *p*(*z*) is sought. Because the boundary conditions at both ends of the model are simply supported ends, for both ends there are f=0 and $f^{\prime \prime }=0$.Then the homogeneous general solution of the governing differential Equation (22) can be obtained as,
(29)$$f(z)=C_{1}+C_{2}z+C_{3}\sinh k_{\xi }z+C_{4}\cosh k_{\xi }z$$
where *C*_1_~*C*_4_ are integration constants. The particular solution $f^{*}(z)$ corresponding to uniformly distributed load can be expressed as,
(30)$$f^{*}(z)=-\frac{pz^{2}}{E_{c}I_{\xi }k_{\xi }^{2}}$$

Based on the boundary conditions at both ends and Equation (14), the generalized moment *M*_ξ_ of simply supported, under uniformly distributed load can be obtained as Equation (31) from Equations (29) and (30):(31)$$M_{\xi }(z)=\frac{p}{k_{\xi }^{2}}\left(\frac{\cosh k_{\xi }l-1}{\sinh k_{\xi }l}\sinh k_{\xi }z-\cosh k_{\xi }z+1\right)$$

Equation (31) shows that *M*_ξ_ of both simply supported ends of the model beam is 0. Then the equivalent distributed loads at both ends of the simply supported beam can be obtained from Equation (27), which are $pe^{-k_{\xi }z}$ and $pe^{k_{\xi }(z-l)}$(as shown in Figure 6). Since the equivalent distributed load decays rapidly as an exponential function, the influence of equivalent distributed load can be ignored at a certain distance from the beam ends. Since the intra-span distributed load *p* does not change with *z*, *M*_ξ_ will not vary with *z*. Thus, in Equation (31) all terms with variable *z* should be 0. Therefore, the approximate relationship between the distributed load *p*(*z*) of CBG-CSWs and the generalized moment *M*_ξ_(*z*) can be obtained as Equation (32) from Equation (31),
(32)$$M_{\xi }(z)=\frac{p(z)}{k_{\xi }^{2}}$$

It can be verified by Equations (32) and (27) that:(33)$$\eta_{2}=\frac{I_{x}}{I_{\xi }k_{\xi }^{2}}$$

Therefore, Equations (27) and (28) are also obtained with the parameters in Equation (33). According to Equations (16) and (32), and the expression of *k*_ξ_, *σ*_ξ_ can be expressed as,
(34)$$\sigma_{\xi }(z)=\frac{E_{c}p(z)}{\lambda \chi G_{w}A_{w}}\xi$$

Similarly, the expressions of warping bimoment B and normal stress $\sigma_{\bar{\omega }}$ of constrained torsion are derived as Equations (35) and (36) from governing differential Equation (23) by the analogy method.
(35)$$B(z)=\frac{-\zeta m(z)}{k_{\varpi }^{2}}$$
(36)$$\sigma_{\bar{\omega }}(z)=\frac{{E^{\prime }}_{c}m(z)}{G_{c}I_{d}}\varpi =\frac{2m(z)}{(1-\mu )I_{d}}\varpi$$
where ϖ is the generalized principal sectorial coordinate [25]; *μ* is the Poisson’s ratio of material. The secondary moment $M_{\varpi }$ can be obtained by taking the derivative of Equation (35), then the constrained secondary shear stress of CBG-CSWs is obtained.

The shear warping stress of statically determinate CBG-CSWs can be obtained by Equations (34)–(36) and the simple material mechanics method. The multi-span continuous beam, commonly used in bridges, can be simplified to several simply supported beams based on the condition that *f* and *φ* at the middle support are 0 and Equation (28). The specific process has been discussed in literature [30] and will not be detailed herein. The simplified method (SM) can quickly solve the shear warping stress of control cross-section of CBG-CSWs’ bridge and facilitate the preliminary analysis of bridge design.

## 4. Beam Segment Finite Element Method

The beam segment finite element method is simpler than the analysis method and three dimensional finite element method (3D-FEM). Moreover, it has a simple modeling process and can directly obtain the internal forces such as bending moment and torsional moment, so it is widely used. The flexural deformation of CBG-CSWs has been decoupled into EBB deformation and shear warping flexural deformation in this paper. There are many studies on the element stiffness matrix of EBB deformation [36], so it will not be detailed herein. Since the governing differential equations of the shear warping flexural deformation and the constrained torsion are consistent in form, the same kind of model can be used to analyze these deformations. The beam segment element model as shown in Figure 7 is established.

In Figure 7, the nodal displacement vectors corresponding to EBB deflection, shear warping deflection and constrained torsion deformation are {***δ***_w_}, {***δ***_f_}, and {***δ****_φ_*}, and the corresponding nodal force vectors are {***F***_w_},{***F***_f_}, and {***F****_φ_*}, respectively.The expressions are as follows:(37a)$$\{\delta_{w}\}={\left[\begin{array}{cccc} w_{0i} & w_{0i}^{\prime } & w_{0j} & w_{0j}^{\prime } \end{array}\right]}^{T}$$
(37b)$$\{\delta_{f}\}={\left[\begin{array}{cccc} f_{i} & f_{i}^{\prime } & f_{j} & f_{j}^{\prime } \end{array}\right]}^{T}$$
(37c)$$\{\delta_{\phi }\}={\left[\begin{array}{cccc} \phi_{i} & \beta_{i} & \phi_{j} & \beta_{j} \end{array}\right]}^{T}$$
(38a)$$\{F_{w}\}={\left[\begin{array}{cccc} Q_{i} & M_{i} & Q_{j} & M_{j} \end{array}\right]}^{T}$$
(38b)$$\{F_{f}\}={\left[\begin{array}{cccc} Q_{\xi i} & M_{\xi i} & Q_{\xi j} & M_{\xi j} \end{array}\right]}^{T}$$
(38c)$$\{F_{\phi }\}={\left[\begin{array}{cccc} T_{\phi i} & B_{i} & T_{\phi j} & B_{j} \end{array}\right]}^{T}$$
where the subscripts *i* and *j* of the vectors are the nodes at both ends of the element; the superscript T is the transpose of the vector; $Q_{i}$ and $Q_{j}$ represent the correspondingshear forces for $w_{0i}$ and $w_{0j}$; *M_i_* and *M_j_* represent the corresponding bending moments for $w_{0i}^{\prime }$ and $w_{0j}^{\prime }$; $Q_{\xi i}$ and $Q_{\xi j}$ represent the corresponding generalized shear forces of shear warping deflections for $f_{i}$ and $f_{j}$; $M_{\xi i}$ and $M_{\xi j}$ represent the corresponding generalized moments for $f_{i}^{\prime }$ and $f_{j}^{\prime }$; $T_{\phi i}$ and $T_{\phi j}$ represent the corresponding torsional moments for $\phi_{i}$ and $\phi_{j}$; $B_{i}$ and $B_{j}$ represent the corresponding warping bimoments of restrained torsion for $\beta_{i}$ and $\beta_{j}$. Obviously, through Equations (37a)–(37c) and (38a)–(38c), the complex deformation sate of CBG-CSWs can be decomposed into three independent deformations, and a similar element stiffness matrix can be used, which simplifies the difficulty of analysis.

### 4.1. Element Stiffness Matrix of CBG-CSWs with Equal Cross-Section

In order to improve the solution accuracy and reduce the computation, the homogeneous solution (29) to differential Equation (22) is used as the displacement function. According to Equation (29), the following expression can be obtained,
(39)$$f^{\prime }(z)=C_{2}+k_{\xi }C_{3}\cosh k_{\xi }z+k_{\xi }C_{4}\sinh k_{\xi }z$$

The shear warping displacements of beam element ends *i* and *j* (*z* = 0 and *z* = *l*) are substituted into Equations (29) and (39) to solve *C*_1_~*C*_4_. Then by substituting the obtained *C*_1_~*C*_4_ into Equation (29), the shear warping displacement *f* of any point of the element can be obtained as,
(40)$$f=\{N_{f}\}\{\delta_{f}\}$$
where $\left\{N_{f}\}=[\begin{array}{cccc} N_{1} & N_{2} & N_{3} & N_{4} \end{array}\right]$ is the generalized warping displacement shape function and the expressions of *N*_1_~*N*_4_ are shown in Appendix A. Substituting Equation (40) into Equation (20) to get,
(41)$$U\;=\;\frac{1}{2}{\left\{\delta_{f}\right\}}^{T}\int_{0}^{l}\left[E_{c}I_{\xi }\left({\left\{N_{f}^{\prime \prime }\right\}}^{T}\{N_{f}^{\prime \prime }\}+k_{\xi }^{2}{\left\{N_{f}^{\prime }\right\}}^{T}\{N_{f}^{\prime }\}\right)\right]dz\{\delta_{f}\}$$

From Equation (41), the stiffness matrix $\left[K_{f}^{e}\right]$ of symmetric element with shear warping deflection can be obtained, and the upper triangular elements of $\left[K_{f}^{e}\right]$ are shown in Appendix B. The equivalent nodal force vector {***F*_f_**} of the element under the external load *p*(*z*) can be obtained according to literature [36].
(42)$$F_{f}=\int_{0}^{l}p(z){\left\{N_{f}\right\}}^{T}dz$$

Since the governing differential Equation (23) has the same form as Equation (22), the interpolation shape function $\left\{N_{\phi }\right\}$ corresponding to torsional displacement can be obtained by the analogue method. Then the element stiffness matrix $\left[K_{\phi }^{e}\right]$ and its load array $\left\{F_{\phi }\right\}$ of constrained torsion deformation can be established. It should be pointed out that the relationship between the torsion angle *φ* and the generalized warping angle *β* should be considered when calculating the torsional stiffness matrix [20,25].

### 4.2. Element Stiffness Matrix of CBG with Varying Depth Cross-Section

In order to reasonably reflect the variation of cross-sections along the length of beam segments, three sections of each element at *i*, *m*, and *j* are selected (as shown in Figure 7). The parameters of the remaining cross-sections are obtained by quadratic polynomial interpolation. From Equation (41), the beam element stiffness matrix $\left[K_{r}^{e}\right]$ considering the variation of cross-section parameters can be obtained:(43)$$[K_{r}^{e}]=\int_{0}^{l}{\left\{N_{r}\right\}}^{T}S(z)\{N_{r}\}dz$$
where *S*(*z*) is the cross-section characteristic parameter; {***N****_r_*} is the first or second derivative of the shape function; for beam segment element with constant cross-section, *S*(*z*) is a constant, while for beam segment element with varying depth cross-section, *S*(*z*) is a variable. Assuming that *S*(*z*) varies along the beam axis in a quadratic parabola, and the cross-section parameters corresponding to *i*, *m*, and *j* of the element are *S_i_*, *S_m_*, and *S_j_*, then:(44)$$S(z)=S_{i}N_{Si}(z)+S_{m}N_{Sm}(z)+S_{j}N_{Sj}(z)$$
where, $N_{Si}(z)=2\frac{z^{2}}{l^{2}}-3\frac{z}{l}+1$; $N_{Sm}(z)=-4\frac{z^{2}}{l^{2}}+4\frac{z}{l}$; $N_{Sj}(z)=2\frac{z^{2}}{l^{2}}-\frac{z}{l}$. By substituting Equation (44) into Equation (43), the stiffness matrix of beam element with varying depth cross-section is obtained:(45)$$[K_{r}^{e}]=[\int_{0}^{l}S_{i}N_{Si}{\left\{N_{r}\right\}}^{T}\{N_{r}\}dz+\int_{0}^{l}S_{m}N_{Sm}{\left\{N_{r}\right\}}^{T}\{N_{r}\}dz+\int_{0}^{l}S_{j}N_{Sj}{\left\{N_{r}\right\}}^{T}\{N_{r}\}dz]\;\;\;\;$$

If the *S_i_*, *S_m_*, and *S_j_* in Equation (45) are set to the cross-section parameters $E_{c}I_{\xi }$ and $E_{c}I_{\xi }k_{\xi }^{2}$, respectively at *i*, *m*, and *j*, and $\left\{N_{r}\right\}$ is the first (second) derivative of $\left\{N_{f}\right\}$, then the element stiffness matrix $\left\{K_{f}^{e}\right\}$ of shear warping deflection of beam with variable cross-section can be obtained; similarly, if *S_i_*, *S_m_*, and *S_j_* are set to the cross-section parameters $E_{c}^{\prime }I_{\bar{\omega }}$ and $E_{c}^{\prime }I_{\bar{\omega }}k_{\bar{\omega }}^{2}$, respectively at *i*, *m*, and *j*, and $\left\{N_{r}\right\}$ is the first (second) derivative of $\left\{N_{\phi }\right\}$, then the element stiffness matrix $\left[K_{\phi }^{e}\right]$ of constrained torsion of beam with variable cross-section can be established; if *S_i_*, *S_m_*, and *S_j_* are respectively set to the cross-section parameter $E_{c}I_{x}$ at *i*, *m*, and *j*, and $\left\{N_{r}\right\}$ is the second derivative of EBB-bending displacement shape function $\left\{N_{w}\right\}$ (Equation (46)) [36], then the bending element stiffness matrix $\left\{K_{w}^{e}\right\}$ of beam with variable cross-section can be obtained.
(46)$$N_{w}=[\begin{array}{cccc} N_{w1} & N_{w2} & N_{w3} & N_{w4} \end{array}]$$
where, $N_{w1}=1-3\frac{z^{2}}{l^{2}}+2\frac{z^{3}}{l^{3}}$; $N_{w2}=z-2\frac{z^{2}}{l}+\frac{z^{3}}{l^{2}}$;

$N_{w3}=3\frac{z^{2}}{l^{2}}-2\frac{z^{3}}{l^{3}}$; $N_{w4}=-\frac{z^{2}}{l}+\frac{z^{3}}{l^{2}}$.

Obviously, when Equation (40) is used as the displacement shape function, the stiffness matrix (45) of beam element and its explicit expression will be extremely complicated. To unify the displacement shape functions of the element, the displacement shape functions of shear warping deflection and constrained torsion deformation of the element adopts Equation (46), i.e.,
(47)$$\left\{N_{f}\}=\{N_{\phi }\}=\{N_{w}\right\}$$

From Equations (45)–(47), the element stiffness matrix of variable cross-section can be derived as,
(48)$$\left[K_{r}^{e}]=\frac{1}{15l^{3}}[K_{1}^{e}]+\frac{n}{420l}[K_{2}^{e}\right]$$
where *n* is 0 or 1. When Equation (48) represents the bending element stiffness matrix of EBB, *n* = 0; when Equation (48) represents the stiffness matrix of shear warping deflection and constrained torsional deformation, *n* = 1. $\left[K_{1}^{e}\right]$ and $\left[K_{2}^{e}\right]$ are both symmetric matrixes and the upper triangular elements of $\left[K_{1}^{e}\right]$ and $\left[K_{2}^{e}\right]$ are shown in Appendix C.

## 5. Numerical Results

The restrained torsional load of CBG-CSWs is applied in the form of eccentric in practical engineering. Under eccentric load, the composite box girder not only has deflection and torsion deformation, but also has distortion deformation. However, the restrained torsion and distortion are coupled together under eccentric load and are difficult to separate. In this paper, the theoretical analysis of the flexure and constrained deformation states is studied without considering the distortion effect. Therefore, the theory proposed in this paper will be verified by establishing a spatial finite element model using the finite software ANSYS. In three-dimensional (3D) finite element, the torsional load is equivalent to the shear flow uniformly applied to the closed box wall element nodes [25].

### 5.1. Example 1: Verification of Simplified Analysis Methods

The CBG-CSWs cross-section shown in Figure 8 is used as an example [10] to verify the solution accuracy of the simplified analysis method, and the calculated span is 2 × 18 m. The load cases are shown in Figure 8b: case 1, the mid-span bears a concentrated load of 340 kN; case 2, the whole bridge bears a uniformly distributed load of 30 kN/m; case 3, the left mid-span bears a concentrated torsional moment of 255 kN·m. The elastic modulus of concrete and steel are 31 GPa and 200 GPa, respectively, and their Poisson’s ratios are 0.2 and 0.3, respectively.

When the simplified analysis method is adopted, the continuous beam can be disconnected from the middle support, and the middle support constraint can be equivalent to the corresponding shear reaction force analysis [30]. The simplified left span analytical model is shown in Figure 9a–c. It can be seen from Equations (32) and (35) that the distribution of shear warping moment of composite box girder with constant cross-section is similar to the distribution of equivalent load. According to the equivalent load distribution in Figure 9, the warping moment of the mid-span and middle-supported cross-section of the composite box girder is large under load cases 1 and 3, while under load case 2 with uniform load distribution, the warping moment of the other cross-sections in the beam span is uniformly distributed except for a certain range of the middle support. For comparative validation, the three-dimensional (3D) finite element model is established by ANSYS. Specifically, the upper and lower flanges of CBG-CSWs adopt soild45 solid elements, CSWs is simulated by the shell63 element, and the shell and solid elements are coupled and connected by MPC (shown in Figure 10).

Through the SM and the initial parameter method (IPM) [20,25], the internal moment of the left mid-span and middle-supported cross-sections of the model beam under load cases 1 to 3 are analyzed, as shown in Table 1. It can be seen that the results obtained by the SM are roughly consistent with those obtained by IPM, but there is a small error at the middle support. The main reason for the error is that the bending moment difference between the equivalent distributed load and the concentrated load is ignored when the SM solves the reaction force of the middle support. From the equivalent load analysis in Figure 9 and the comparative results in Table 1, it shows that the SM is reliable in the whole beam range for the composite box girder with constant cross-section.

Figure 11 and Figure 12 show the vertical distribution of the flexural stress of the cross-sections near the mid-span and middle support of the CBG-CSWs under load cases 1 and 2, respectively. In the figures, *η*_1_ and *η*_2_ are the coefficients obtained by calculating the shear force in different ways when the SM is used (Section 3.2). It can be observed from Figure 11 and Figure 12 that the stress values obtained by the SM are in good agreement with the 3D FEM results. In Figure 11a, the stress analysis result of EBB on the top of the upper flange is approximately −0.5 MPa, while the 3D FEM result is −0.94 MPa. It is shown that the EBB [6,7,8] stress analysis results, without considering the effect of shear deformation, are quite different from 3D FEM at the top and bottom of the flange. Compared with the EBB analysis results, the traditional Timoshenko theory can only improve the calculation’s accuracy of deflection. Therefore, it is difficult to obtain the more accurate stress distribution of the composite box girder by using the Timoshenko beam theory. The proposed SM can not only obtain more accurate stress, but also be more convenient for calculation.

To further investigate the impact of shear warping deformation, Figure 13 shows the curves of the flexural stress at the top of the upper flange along the beam axis under load cases 1 and 2. Figure 13 shows that the analysis results considering shear warping deformation are in good agreement with 3D FEM, indicating a high calculation accuracy of the SM. Compared with the bending stress results of EBB, the influence range of shear warping deformation is mainly distributed in the cross-section near the middle support and the concentrated load. It further shows that the SM can quickly and accurately obtain the shear warping stress of the control cross-section of CBG-CSWs, which is suitable for application in engineering projects. Figure 14 shows the distribution of the torsional warping normal stress along the beam axis at point 1 of the cross-section (shown in Figure 8a) under load case 3. It can be seen from Figure 14 that the results of SM correspond well with the analytical results of IPM and differ slightly with 3D FEM results. The reason for this difference may be that the 3D FEM results contain the distortion warping stress of the CBG-CSWs.

Table 2 lists the displacement results calculated by different methods. As shown in Table 2, the displacement results calculated by the SM under each load case are in good agreement with the IPM and 3D FEM analyses results. Under load cases 1 and 2, the deflection increment caused by shear warping deformation exceeds the deflection calculated by EBB. After considering the influence of shear warping deformation, the deflection is smaller than the result calculated by 3D FEM, because the increase in deflection caused by shear lag is not considered [30].

### 5.2. Example 2: Precision Analysis of Displacement Shape Function

In Section 4, {***N***_f_} and {***N***_w_} are respectively used as displacement shape functions to obtain the stiffness matrix of the beam segment element. In order to verify the solution accuracy of the two displacement shape functions, the analysis programs FBOX and WBOX are developed respectively based on the two element stiffness matrices.

The continuous beam in Section 5.1 is still used as the analytical model. The programs FBOX and WBOX are respectively applied to analyze the stress and displacement of CBG-CSWs mid-span cross-section under load cases 1 and 2. The analytical results are compared with analytical results and normalized. Figure 15 shows the convergence curve of the normalized results with the increasing number of uniformly divided elements.

It can be seen from Figure 15 that the stress and displacement results of FBOX analysis converge to the analytical solution under concentrated load, which are independent of the number of elements. When subjected to uniformly distributed load, the convergence rates of the FBOX and WBOX analysis programs are roughly the same, indicating that both programs have high efficiency and accuracy.

### 5.3. Example 3: Verification of Beam Segment Finite Element Method for Varying Depth Cross-Section

Based on the finite element model with varying cross-section depth in Section 4.2, the beam segment finite element analysis program VWBOX is compiled. To verify the efficiency and accuracy of VWBOX, the continuous CBG-CSWs with varying cross-section depth (shown in Figure 16) is selected for calculation. The total length of CBG-CSWs is 6150 mm, and the calculated span is 2 × 3000 mm [37]. The cross-section sizes of the middle support (A-A) and side support (D-D) of the beam with variable cross-sections and the CSW parameters are shown in Figure 16b. The elastic modulus of concrete is 34.5 GPa and its Poisson’s ratio is 0.2. The elastic modulus of steel is 206 GPa and its Poisson’s ratio is 0.3.

Considering the corner haunch of concrete flanges in actual bridge construction, when calculating the cross-section parameters, the center lines of upper and lower concrete flanges are selected as the centroid of each flange to obtain *h*_u_ and *h*_l_. When the beam segment element program is used for analysis, the calculated span of the whole bridge is divided into 60 equal-length elements along the beam axis. The variable section beam segment program (VWBOX) and constant section beam segment program (CWBOX) compared with the SM. Table 3 shows the internal moment results of cross-sections A-A and C-C. It can be seen from Table 3 that the internal forces of control cross-sections obtained by the two programs are quite close. There is little difference between the generalized internal forces of the control cross-section obtained by SM and the analysis program results, which further indicates the reliability of SM. The distribution of the normal stress at the top of the upper flange along the beam axis is shown in Figure 17, where the analysis results of 3D FEM are presented for comparison. It can be observed that the results of VWBOX are in good agreement with the stress calculated by 3D FEM, indicating the effectiveness of the program. The advantage of this program is that it can obtain both the shear warping deformation results and the EBB deflection analysis results, which are suitable for application.

For CBG-CSWs with varying depth cross-section, shear stress of CSWs has always been the focus of relevant research [34,35]. *f*′(*z*) can be easily obtained by VWBOX, and then the flexural shear stress *τ*_w_ of CSWs can be obtained by Equation (19). The web shear stress obtained by this method is feasible for CBG-CSWs with constant-section. However, for CBG-CSWs with varying cross-section parameters such as centroid axis and beam height, the influence of these parameters on shear stress should be considered [34,35]. The cross-section parameters and bending moment *M* can be used to modify the web shear stress, and the specific method can be found in the literature [34,35]. Figure 18 shows the vertical distribution of flexural shear stress of cross-section B-B, where CS and UCS represent the corrected and uncorrected shear stresses, respectively. Obviously, the modified CSWs shear stress is consistent with 3D-FEM result, which further verifies the accuracy of the VWBOX.

## 6. Conclusions

This paper presents a new practical method for the analysis of Flexural and restrained torsional deformations of CBG-CSWs. By introducing shear warping deflection and corresponding internal force, flexural deformation is decoupled into two deformations: EBB deformation and shear warping deflection. From the root cause of shear warping deflection, a SM of shear warping deflection is proposed based on equivalent shear conditions. According to the similarity of the governing differential equations of constrained torsion and shear warping deflection, a SM for constrained torsion of CBG-CSWs is derived. Numerical examples of two continuous CBG-CSWs with constant cross-section and variable cross-section demonstrate the effectiveness of the SM. Based on the decoupled deformation states, the EBB flexural deformation, shear warping deflection and constrained torsion deformation are unified into the element stiffness matrix with the same form. A variable-section beam segment analysis program is developed. Comparative analysis of numerical examples of constant- and variable-section CBG-CSWs shows that:(1)Compared with other methods, the decoupled shear warping deflection is simpler to analyze and its physical significance is more clear.(2)The stress and deformation analytical results of numerical examples show that the SM and beam segment program proposed in this paper are effective and have high efficiencies when applied in practical engineering projects.(3)When other influencing factors such as diaphragm plates are ignored, shear warping deformation has a great influence near the concentrated load and the middle support. The influence range of shear warping deformation along the beam axis decays exponentially and the attenuation rate is related to the shear warping coefficients ($k_{\xi }$ and $k_{\bar{\omega }}$) of section.(4)The deflection caused by shear warping deformation exceeds that by EBB. In the cross-section near the middle support and the concentrated load, the stress increase at the top of the upper flange caused by shear warping deformation is larger than the results from EBB analysis.

## Figures and Tables

**Figure 1 materials-16-01845-f001:**
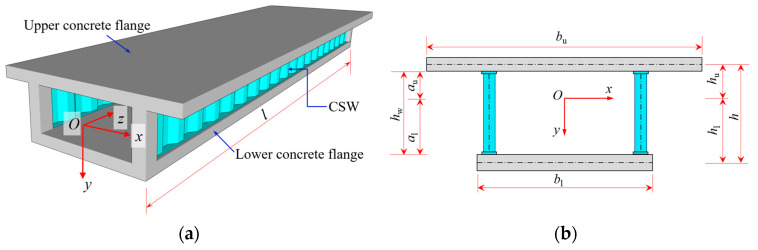
A typical CBG-CSWs modal: (**a**) coordinate system; (**b**) cross-section.

**Figure 2 materials-16-01845-f002:**
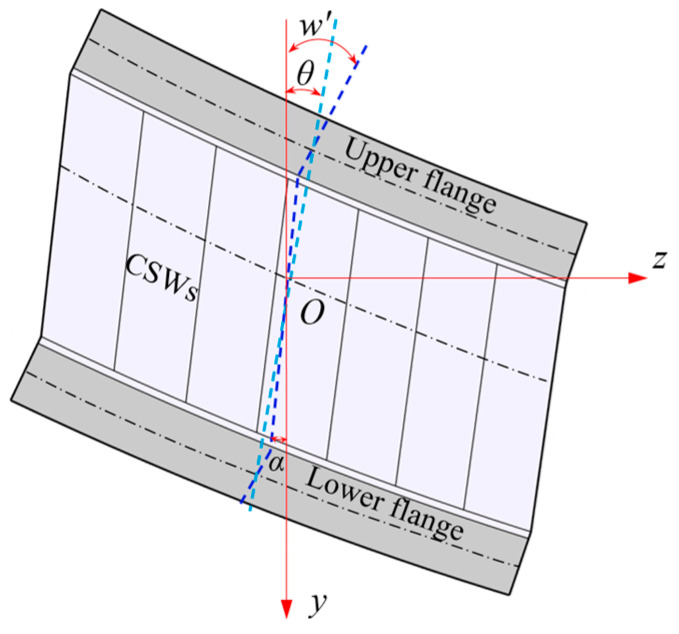
Vertical flexural deformation of CBG-CSWs.

**Figure 3 materials-16-01845-f003:**
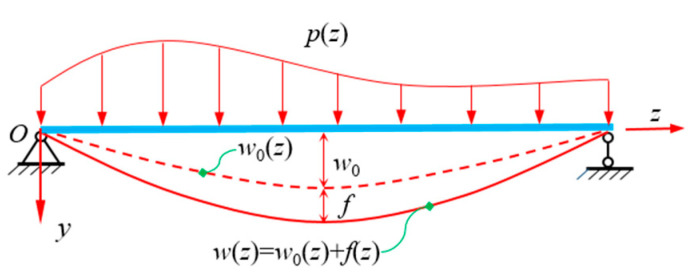
Decoupling of vertical deflection.

**Figure 4 materials-16-01845-f004:**
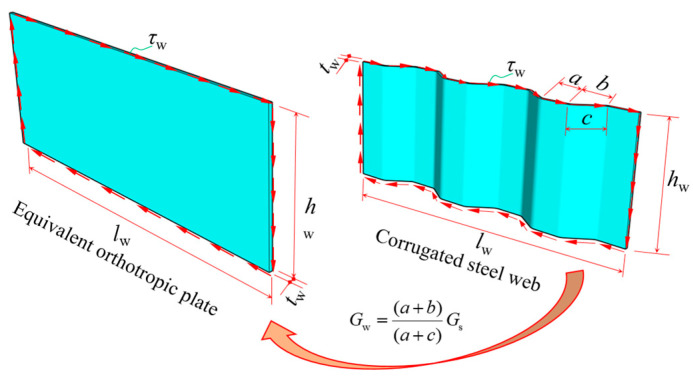
Equivalent of corrugated steel webs.

**Figure 5 materials-16-01845-f005:**
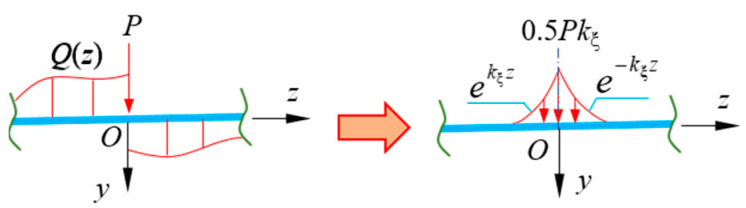
Equivalent analysis of shear discontinuities.

**Figure 6 materials-16-01845-f006:**
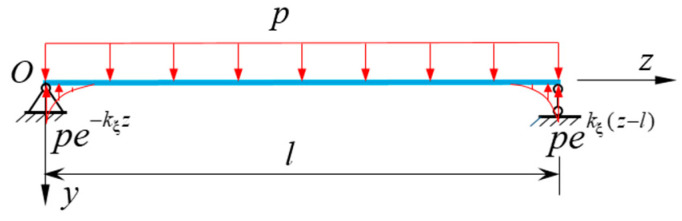
Asimply supported beam.

**Figure 7 materials-16-01845-f007:**
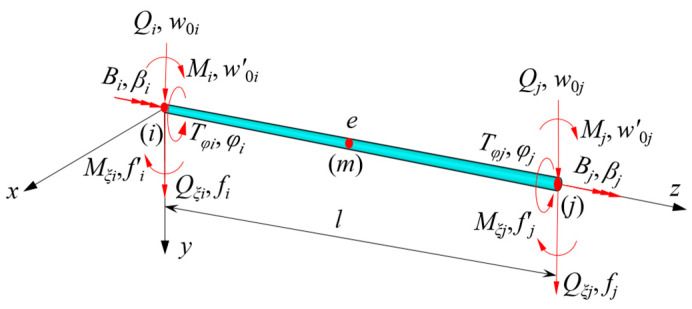
Beam-type finite element model.

**Figure 8 materials-16-01845-f008:**
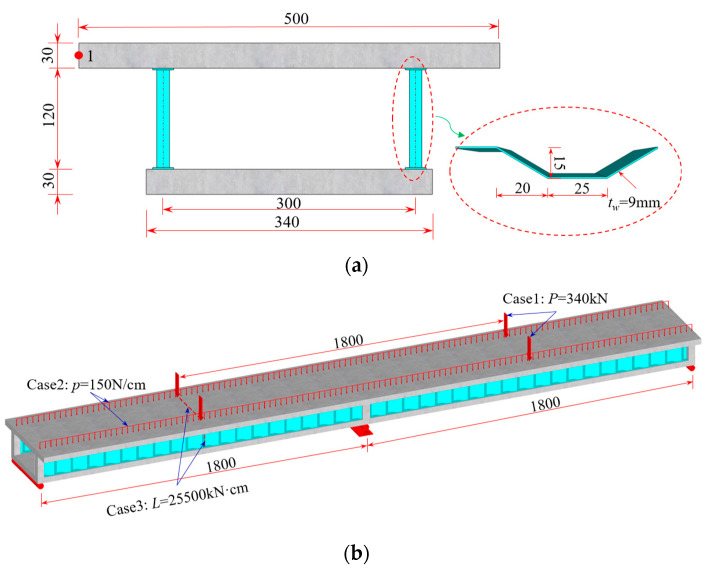
Two span continuous CBG-CSWs (unit: cm): (**a**) cross-section; (**b**) load cases.

**Figure 9 materials-16-01845-f009:**
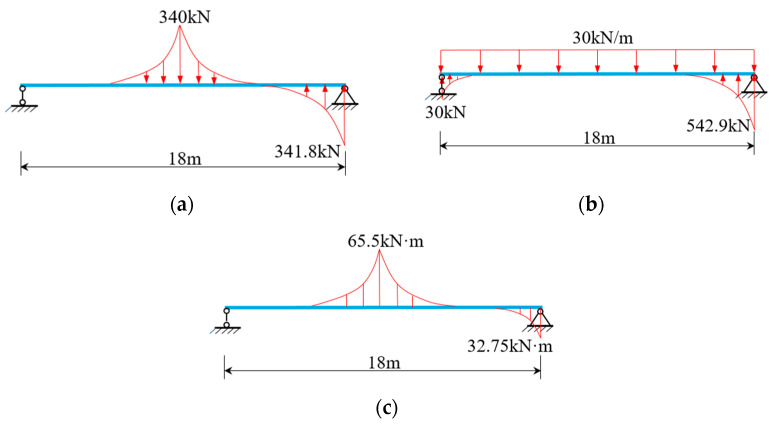
Equivalent distributed load of two-span continuous beam under different load cases: (**a**) Case 1; (**b**) Case 2; (**c**) Case 3.

**Figure 10 materials-16-01845-f010:**
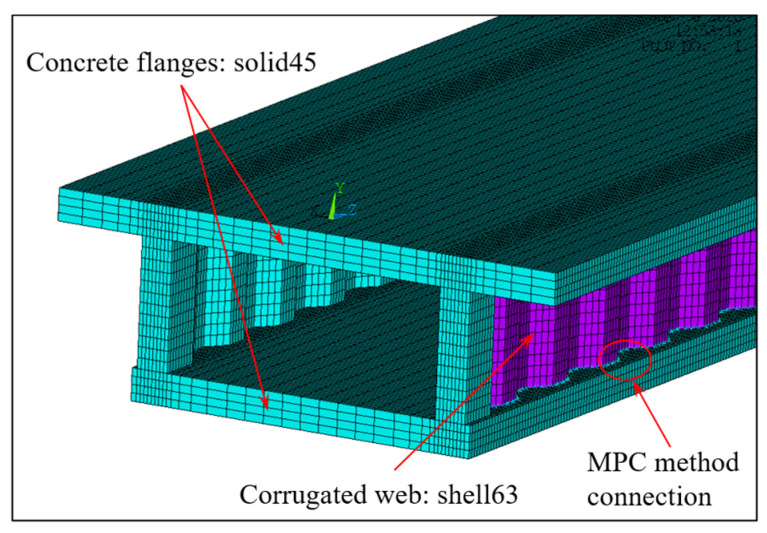
3D finite element model of CBG-CSWs.

**Figure 11 materials-16-01845-f011:**
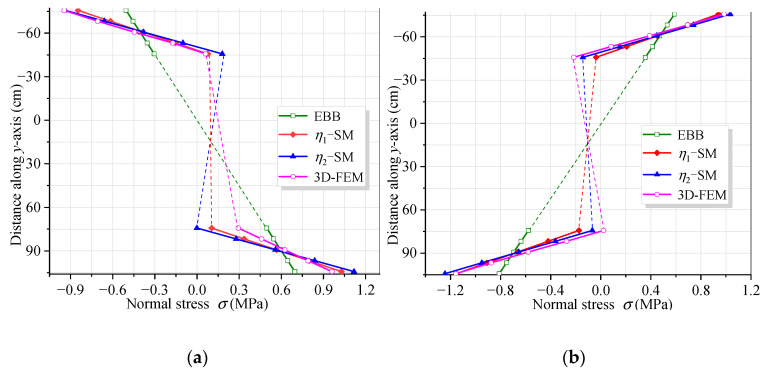
Vertical distribution of normal stress along the cross-section under load case 1: (**a**) cross-section 9.3 m away from the middle support; (**b**) cross-section 0.3 m away from the middle support.

**Figure 12 materials-16-01845-f012:**
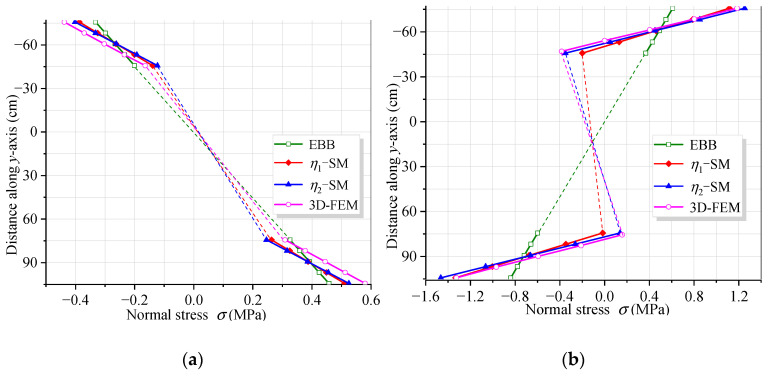
Vertical distribution of normal stress along the cross-section under load case 2: (**a**) cross-section 9.0 m away from the middle support; (**b**) cross-section 0.3 m away from the middle support.

**Figure 13 materials-16-01845-f013:**
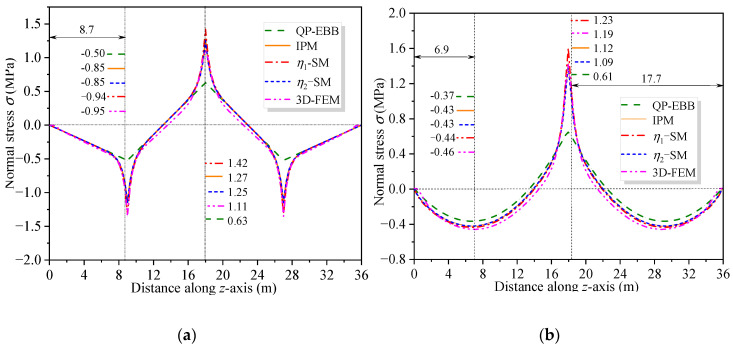
Distribution curve of normal stress at the flange top along the beam axis: (**a**) Case 1; (**b**) Case 2.

**Figure 14 materials-16-01845-f014:**
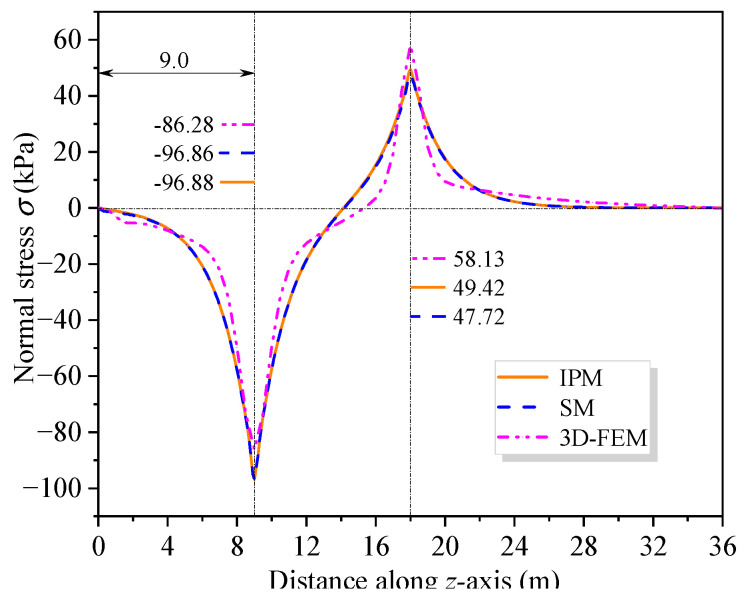
Distribution curve of flange torsional warping normal stress along the beam axis.

**Figure 15 materials-16-01845-f015:**
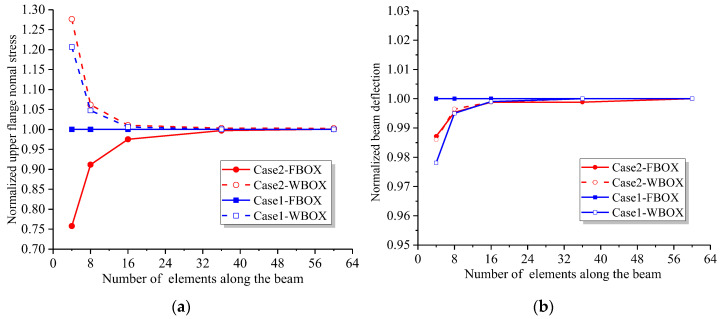
Convergence analyses of the FBOX and WBOX elements: (**a**) stress convergence curve; (**b**) deflection convergence curve.

**Figure 16 materials-16-01845-f016:**
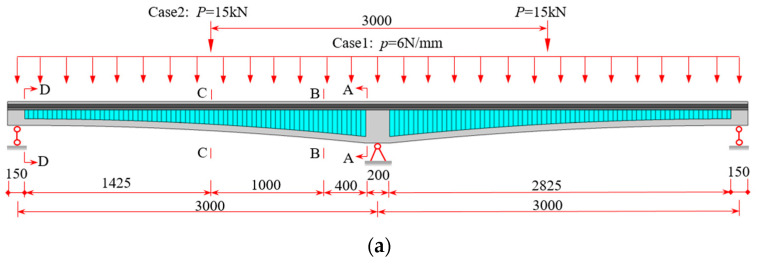
Two-span continuous CBG-CSWs (unit: mm): (**a**) load cases layout; (**b**) cross-section.

**Figure 17 materials-16-01845-f017:**
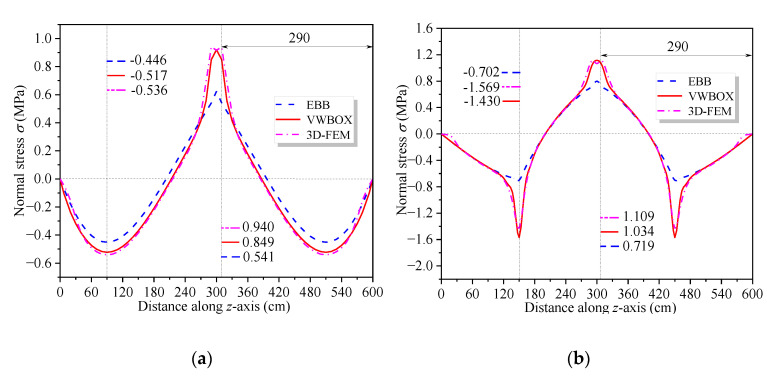
Normal stress distribution along the beam axis at the top of the upper flange under different load cases: (**a**) Case 1; (**b**) Case 2.

**Figure 18 materials-16-01845-f018:**
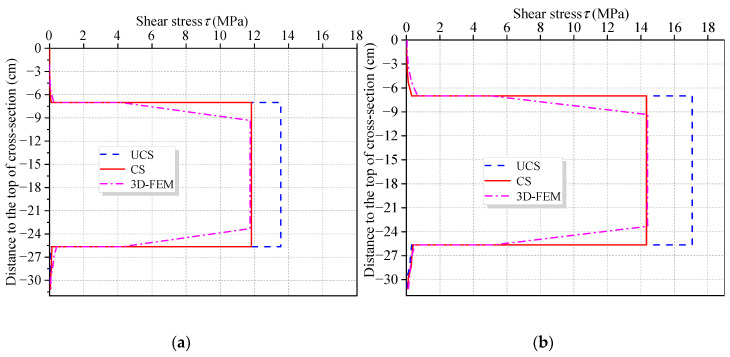
Vertical distribution of shear stress along section B-B under different load cases: (**a**) Case 1; (**b**) Case 2.

**Table 1 materials-16-01845-t001:** Moment in the mid-span and middle-supported cross-section of CBG-CSWs.

Load Cases	Mid-Span Cross-Section			Middle-Supported Cross-Section		
	SM	IPM	Error (%)	SM	IPM	Error (%)
Case 1 *M*_ξ_ (kN·m)	84.55	84.55	0	−84.55	−86.95	−2.8
Case 2 *M*_ξ_ (kN·m)	7.42	7.42	0	−126.86	−129.85	−2.3
Case 3 *B* (kN·m^2^)	78.16	78.14	0.02	−38.50	−39.86	−3.4

Note: Error = (SM − IPM)/IPM × 100.

**Table 2 materials-16-01845-t002:** Displacement results of the mid-span cross-section.

Method	Case 1				Case 2				Case 3	
	*w*_0_ (mm)	*f* (mm)	*w* (mm)	Error (%)	*w*_0_ (mm)	*f* (mm)	*w* (mm)	Error (%)	*φ* × 10^−3^ (rad)	Error (%)
EBB	−0.421	--	−0.421	61.7	−0.382	--	−0.382	60.7	--	--
IPM	−0.421	−0.584	−1.005	8.6	−0.382	−0.476	−0.858	11.7	0.076	−1.3
SM	−0.421	−0.581	−1.025	6.8	−0.382	−0.476	−0.858	11.7	0.071	5.3
3D FEM	--	--	−1.100	0	--	--	−0.972	0	0.075	0

Note: error is the difference between the 3D FEM analysis result and the calculation result of other methods divided by the 3D FEM analysis result.

**Table 3 materials-16-01845-t003:** Internal moment of mid-span and middle support of CBG-CSWs (uint: kN·cm).

Load Cases	A-A			C-C		
	SM	VWBOX	CWBOX	SM	VWBOX	CWBOX
Case 1 *M*_ξ_	−25.24	−26.62	−25.56	6.50	6.50	7.48
Case 1 *M*	--	−752.36	−752.37	--	240.85	240.85
Case 2 *M*_ξ_	−26.09	−27.19	−27.13	78.34	78.30	78.23
Case 2 *M*	--	1112.85	1112.86	--	568.58	568.57

## Data Availability

The data presented in this study are available on request from the corresponding author.

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
