# Peer review of "Study of Practical Analysis Method for Shear Warping Deformationof Composite Box Girder with Corrugated Steel Webs"

_materials, 2023, doi:10.3390/ma16051845_

Round 1
Reviewer 1 Report
Please see the attached file.

Author Response
Dear Editor and Reviewers
Thank you for your letter and for the reviewers’ comments and suggestions concerning our previous version of the manuscript. Those comments are all valuable and very helpful for revising and improving our paper, as well as the important guiding significance to our researches. We have studied comments carefully and have made corrections which we hope meet with approval.
We have uploaded the file of the revised manuscript and a copy of the original manuscript with all the changes highlighted by using the revision mode.

Reviewer 2 Report
In this manuscript, the authors have studied shear warping deformation using finite element methods (Ansys). In overall, manuscript was quite technical and motivation for this study is hard to find. Also the novelty of the study stays questionable, there are number of similar FEM studies close to this topic and authors have not be clear enough to explain how this study have not done before by other authors. However, I did not find mistakes from the equations and there is enough detailed representation to reproduce this study, so with these additions (and rewriting main parts of the text more fluent manner) I recommend major revision to this manuscript.
Author Response

(The authors gave the same response as above.)

Reviewer 3 Report
A large amount of research has been carried out, but all research is focused only on process modeling.
1. To what extent will the simulation data coincide with the real physical experiment?
2. Will this mechanism work on all materials?
Author Response

(The authors gave the same response as above.)

Reviewer 4 Report
This research works proposed a new practical technique for the analysis of Flexural and restrained torsional deformations of CBG-CSWs. The paper can be accepted after the revision.
- Please add a notation list.
-In order to provide a more comprehensive literature review, the authors should cite and discuss the following relevant papers in their revised manuscript:
SCFs in tubular X-connections retrofitted with FRP under in-plane bending load. Composite Structures, 274, 2021;p.114314.
Qiao, P., Di, J. and Qin, F.J., 2018. Warping torsional and distortional stress of composited box girder with corrugated steel webs. Mathematical Problems in Engineering, 2018.
- Add sensitive analyze of the mesh size.
- In lines 69 and 70, “Cambronero-Barrientos [25] et al. proposed” should be “Cambronero-Barrientos et al. [25] proposed”. Alos, in line 239, “Eq. (16) and (27)” should be “Eqs. (16) and (27)”
- Figs. 11a and 11b need more discussion in the text.
Author Response

(The authors gave the same response as above.)

Round 2
Reviewer 2 Report
It's ok now.
Reviewer 4 Report
The paper can be accepted.